# Optimised degradation correction for SCIAMACHY satellite solar measurements from 330 to 1600 nm by using its internal white light source

Tina Hilbig[1], Klaus Bramstedt[1], Mark Weber[1], John P. Burrows[1], and Matthijs Krijger[2]

[1]Institute of Environmental Physics, University of Bremen, Bremen, Germany
[2]Earth Space Solutions, Utrecht, The Netherlands

**Correspondence:** hilbig@iup.physik.uni-bremen.de, klaus.bramstedt@uni-bremen.de

**Abstract.** SCIAMACHY (SCanning Imaging Absorption spectroMeter for Atmospheric CHartographY) on-board the European Environmental Satellite (Envisat) provided spectrally resolved measurements in the wavelength range from 0.24 to 2.4 $\mu$m by looking into the Earth's atmosphere using different viewing geometries (limb, nadir, solar and lunar occultation). These observations were used to derive a multitude of parameters, in particular atmospheric trace gas amounts. In addition to radiance measurements solar spectral irradiances (SSI) were measured on a daily basis. The instrument was operating for nearly a decade, from August 2002 to April 2012. Due to the harsh space environment, it suffered from continuous optical degradation. As part of recent radiometric calibration activities an optical (physical) model was introduced that describes the behaviour of the scanner unit of SCIAMACHY with time (Krijger et al., 2014). This model approach accounts for optical degradation by assuming contamination layers on optical surfaces in the scanner unit. The variation of layer thicknesses of the various optical components is determined from the combination of solar measurements from different monitoring light paths available for SCIAMACHY. In this paper, we present an optimisation of this degradation correction approach, which in particular improves the solar spectral data. An essential part of the modification is the use of measurements from SCIAMACHY's internal white light source (WLS) in combination with direct solar measurements. The WLS, as an independent light source, gives, therefore, an opportunity to better separate instrument variations and natural solar variability. However, the WLS emission depends on its burning time and is changing with time as well. To use these measurements in the optimised degradation correction, the change of the WLS emission in space needs to be characterised first. The changes of the WLS with accumulated burning time are in good agreement with detailed laboratory lamp studies by Sperling et al. (1996). Although the optimised degradation corrected SCIAMACHY SSI show still some instrumental issues when compared to SSI measurements from other instruments and model reconstructions, our study demonstrates the potential for the use of an internal WLS for degradation monitoring.

## 1 Introduction

In order to understand the Earth's climate and its changes the Earth's atmosphere is studied by numerous instruments from space since the 1970s. Only space-based measurements have the potential to obtain geophysical parameters nearly simultaneously on a global scale and over long periods. To retrieve the content and vertical profiles of trace gases in the atmosphere the UV

and visible radiance backscattered from the Earth are normalised by the incoming solar radiation (e.g. Bovensmann et al., 2011). Therefore, atmospheric sounders operating in the optical spectral range provide top of the atmosphere solar spectral irradiance (SSI) measurements, which are often performed on a daily basis. They provide a large data base with the potential for solar studies as well. Investigations of SSI are essential since the Sun is the primary energy source of the Earth's climate with a contribution of about 99.96% (Kren et al., 2017). The Sun's radiative output changes with the level of solar activity. These changes impact the chemical and dynamical processes in the Earth's atmosphere (such as ozone chemistry, temperature structure) and therefore the climate (e.g. Haigh, 2007; Gray et al., 2010; Solanki et al., 2013; Lean, 2017). Moreover, solar irradiance variations show a strong wavelength dependence with the highest absolute irradiance change in the visible and near ultraviolet (NUV) spectral range, but the largest fractional change in the ultraviolet (UV) and below (e.g. Krivova et al., 2009; Ermolli et al., 2013; Lean, 2017). To account for the influence of incoming solar radiation on Earth's atmosphere and climate, accurate knowledge of the absolute solar radiation for the whole wavelength range as well as its relative changes with time (e.g. 11-year solar cycle, 27-day solar rotation) is required. While the absolute accuracy of SSI measurements are a few percent (1 to 3%), detection of variations in SSI require a measurement precision in the sub-percentage range.

Due to the absorption of solar UV radiation and IR bands in the atmosphere, space-based measurements are the only possibility for direct Sun observations. Regular space-based measurements of the solar irradiance begun in 1978 (Ermolli et al., 2013) with the *Solar Backscatter UltraViolet instrument* (SBUV/-2) on Nimbus-7 and the NOAA satellites. The first observations focused on the UV (below 400 nm) where variability is strongest over the 11-year solar cycle (e.g. DeLand et al., 2012; Woods et al., 2018). The *Upper Atmosphere Research Satellite* (UARS) (Reber et al., 1993) was carrying instruments to sound the upper atmosphere as well as two instruments to measure the SSI from 400 nm down to 120 nm (the *Solar Ultraviolet Spectral Irradiance Monitor*, SUSIM, and the *Solar Stellar Intercomparison Experiment*, SOLSTICE) between 1991 – 2005 and contributed therefore to many atmospheric long-term data records (e.g. Floyd et al., 2003).

SCIAMACHY (Burrows et al., 1995; Gottwald et al., 2011; Pagaran et al., 2009, 2011) was one of the first satellite spectrometers to measure over the entire optical wavelength range (UV to shortwave-infrared) on a daily basis from 2002 until early 2012. SCIAMACHY was part of a series of instruments with spectral coverage up to about 800 nm: GOME, the *Global Ozone Monitoring Experiment*, (Burrows et al., 1999) aboard ERS-2 (1995 – 2011) and the GOME-2 follow-up instruments (Munro et al., 2016) aboard EUMETSAT's Metop satellite series since 2007.

Shortly after SCIAMACHY was launched, the *Spectral Irradiance Monitor* (SIM, 240–2400 nm, Harder et al. (2005)) and the *Solar Stellar Comparison Experiment* (SOLSTICE, 116–310 nm, McClintock et al. (2005)) on-board the SORCE (*Solar Radiation and Climate Experiment*) satellite started their operation in 2003. An intensive scientific debate started when first results of the SORCE satellite were published by Harder et al. (2009); Haigh et al. (2010). They showed a four to six times larger solar UV variability over the 11-year solar cycle than what was known before from observations and models (e.g. DeLand et al., 2012; Ermolli et al., 2013). These results implied an anti-correlation of the solar irradiances in the visible and the 11-year solar cycle, which is inconsistent with other satellite measurements that show in-phase variations (Wehrli et al., 2013; Marchenko and DeLand, 2014). Recent studies (Mauceri et al., 2018; Woods et al., 2018, 2015) indicate that the UV variations of SORCE/SIM were initially overestimated and these solar irradiance changes are consistent with possible

instrument sensitivity drifts (Lean and DeLand, 2012). Chemistry-climate model simulations show that different input data sets might have significant influences on the responses and therefore implications of Earth's atmosphere (*e.g.* Ermolli et al., 2013; Wen et al., 2017). The time series of SIM is continued by an improved SIM instrument as part of NASA's *Total and Spectral Solar Irradiance Sensor* (TSIS-1, since late 2017) mounted at the International Space Station (ISS) (Pilewskie et al., 2018) and in a compact version on CubeSat as *Compact Spectral Irradiance Monitor* (CSIM) launched in December 2018 (Harber, 2019).

Another data record with extensive wavelength coverage in the optical spectral range is available from the *SOLar SPEC-trometer* (SOLSPEC) as part of the *Solar Monitoring Observatory* (SOLAR) payload on the ISS (e.g. Thuillier et al., 2009; Meftah et al., 2018). SOLSPEC measured the solar irradiance from 2008 to 2017 in the wavelength range 165 – 3000 nm. Further regular measurements in the UV to vis (265 – 500 nm) were provided by the *Ozone Monitoring Instrument* (OMI, since 2004) on-board the *AURA* satellite (Levelt et al., 2006; Marchenko and DeLand, 2014) and its follow-up *TROPOspheric Monitoring Instrument* (TROPOMI, since 2017) on-board the Copernicus Sentinel-5 Precursor (S5P) (Veefkind et al., 2012).

One challenge for space-based measurements is the harsh environment of operation (vacuum, high energetic particles, temperature extremes, ultraviolet radiation, etc.). Among others, this environment can cause relatively rapid degradation of the satellite instruments which becomes problematic in particular when measuring over long periods (e.g. DeLand et al., 2012; Lean and DeLand, 2012; Morrill et al., 2014). More precisely, space-based instruments suffer from instrumental artefacts that lead to deposits of contaminants on optical surfaces (e.g. Krijger et al., 2014). Contaminants can originate from outgassing or evaporation by all organic material used in the construction of these instruments (BenMoussa et al., 2013). Another critical contaminant is water that can be deposited on cooled surfaces in the instrument as it was the case for SCIAMACHY's NIR detectors Lichtenberg et al. (2006). In addition, water vapour is photolysed in the hard UV to generate OH and H free radicals. In particular OH is a very strong oxidising agent and initiates the oxidation of volatile organic compounds which are also photolysed in the UV and result in additional low volatile organic compounds deposited on the optical surfaces of the instrument Meftah et al. (2017) provided an extensive list of possible contamination sources, like machining oils, cleaning solvents, bagging material, propulsion systems, attitude and orbit control systems, materials outgassing, etc. as well as possible stage of contamination, e.g. fabrication, assembly, purges, venting, test, storage, transport, launch site and ascent, spacecraft separation from the launcher and manoeuvres or the on-orbit commissioning phase. The mechanisms of instrumental degradation in space are complex and in most cases a combination of several independent processes (BenMoussa et al., 2013). In summary, the main reason for instrument degradation seems to be a combination of the presence of contaminant species and the exposure to solar radiation. The UV radiation can cause polymerisation of organic material and, subsequently, irreversible deposition of this material. This leads to changing, mainly growing, contamination layers on the instruments' optical surfaces (Krijger et al., 2014) and changes in instrument response with time (e.g. Morrill et al., 2014). Detailed discussions on contamination of optical surfaces in various space instruments are provided by BenMoussa et al. (2013); Krijger et al. (2014); Meftah et al. (2017) among others. Despite the numerous studies available, the composition of contaminants as well as the exact processes for their build-up remain highly uncertain. For each instrument and platform, the individual construction (platform, instrument) and performance lead to different effects, and are difficult to quantify without having direct access in space. SCIAMACHY and

its precursor GOME show moderate degradation in the UV and visible spectral range, whereas the successor GOME-2 series show more rapid degradation.

Therefore, one challenge of spectral solar measurements from space is the development of a thorough degradation correction to assess and maintain instrument calibration over the entire instrument lifetime. In the past a few methods have been developed to track in-flight instrument degradation (Woods et al., 2018, e.g.). The first implies the integration of redundant instruments (or components). Different operation times (or exposure times) in the redundant components provide information on the degradation correction. This was done with the SUSIM instrument using redundant calibration lamps and the SIM with a redundant spectrometer (Harder et al., 2005, 2010). The second option is the use of external calibration sources like viewing stable UV stars as in the case of the SOLSTICE instruments aboard UARS and SORCE (McClintock et al., 2005). A third type of in-flight calibration uses measurements with on-board calibration lamps (stable light source) to monitor instrument degradation. Despite many attempts to eliminate degradation effects, e.g. extreme cleanliness control, minimisation of organic material, as well as careful initial calibration and monitoring of instrument calibration in orbit, uncertainties remain in the time series and resulting trends (e.g. Ermolli et al., 2013).

The goal of our investigations is to improve the absolute radiometric calibration of the time series of the solar extraterrestrial spectra measured by SCIAMACHY. The degradation is derived relative to a reference measurement with published absolute calibration (Hilbig et al., 2018). This is achieved by adaption of the degradation correction for the SCIAMACHY instrument by modifying the mirror model approach (Krijger et al., 2014) as described in Section 3. The strategy is to use measurements of SCIAMACHY's internal white light source, as an independent light source in combination with solar monitoring measurements to derive the degradation correction. Before presenting the improved degradation correction, a brief description of the SCIAMACHY solar measurements and instrument operation is provided in Section 2. The re-calibrated SCIAMACHY SSI are presented in Section 4. In Section 5 we compared SCIAMACHY SSI time series with other solar data and investigate the quality of the re-calibrated SCIAMACHY SSI time series.

## 2   SCIAMACHY measurements from 2002-2012

A brief summary of the SCIAMACHY instrument and its solar measurements is provided in this section. A more detailed description is given in Hilbig et al. (2018) with a particular focus on the evolution of the radiometric calibration with the ESA (*European Space Agency*) processing versions of the Level 1 data (solar irradiance, backscattered radiances) up to version 9.01[1]. However, the version 9.01 data will not be released to the public as its degradation correction introduced a small and unexpected long-term trend in the retrieved total ozone. This work thus contributes to further improvements to the degradation corrections going beyond ESA V9.01.

---

[1]Version 9 in Hilbig et al. (2018) is identical with V9.01.

## 2.1 Instrument

SCIAMACHY was one of ten remote sensing instruments on board ESA's Envisat satellite platform. It was constructed primarily to study trace gases in the terrestrial atmosphere (Burrows et al., 1995; Gottwald and Bovensmann, 2011). To normalise the radiance backscattered from the Earth, the incoming solar radiation was measured on a daily basis.

The instrument comprises a scan mirror system, a telescope, and a spectrometer, controlled by thermal and electronic subsystems. The scan mirror system consists of the elevation (ESM) and azimuth scan mechanisms (ASM) that permits the various viewing geometries (nadir, limb, solar and lunar occultation, and direct solar and lunar measurements), as shown schematically in Figure 1. The spectrometer disperses the solar radiance/irradiance into eight spectral bands (channels) covering the UV-vis-NIR-SWIR. It is a double monochromator, consisting of a pre-disperser prism and gratings. Each spectral channel has its own grating and detector.

## 2.2 Solar Measurements

Solar observations were made by using different combinations of scan mirrors (elevation and azimuth scan mirrors, ESM and ASM) and diffusers which were mounted on the backside of the mirrors. The so-called ESM diffuser measurements are the only pre-flight absolutely radiometrically calibrated solar measurements and provide the solar spectra in physical units. Regular solar observations (via ESM diffuser) were performed once a day in a measurement sequence of 30s. The spectral range covered is from 212 to 1773 nm and two narrow bands from 1934 to 2044 nm and 2259 to 2386 nm at a moderate spectral resolution of 0.2 – 1.5 nm (Hilbig et al., 2018).

## 2.3 Calibration

The calibration of SCIAMACHY solar measurements includes the following steps (Slijkhuis and Lichtenberg, 2018): Detector corrections address the pixel-to-pixel gain (diode arrays), the memory effect in the UV and visible channels (residual signal from the previous detector readout), and the nonlinearity effect of the NIR detectors. Dark signal corrections are based on dark current measurements performed during night time of each orbit. Furthermore, a stray light correction and polarisation correction that accounts for the polarisation sensitivity of the instrument were applied (Lichtenberg et al., 2006). SCIAMACHY's spectral calibration uses in-flight measurements of atomic spectral lines from the internal Spectral Line Source (SLS), a Pt/Cr-Ne hollow cathode lamp (Slijkhuis and Lichtenberg, 2014). The absolute radiometric calibration uses on-ground calibrations that were carried out pre-flight using a combination of spectralon/NASA sphere and FEL lamps (Filament Emission Lamp). The radiometric calibration accounted for the scan angle dependence of the signal, since the reflectivity of mirrors and diffusers changes with the incidence angle. Changes of calibration parameters due to the transition of the instrument from pre-flight to in-orbit conditions were adjusted by an on-ground to in-flight correction, as described in Hilbig et al. (2018). A degradation correction determined using in-flight measurements is applied to account for instrument throughput changes during the mission lifetime (see Section 3).

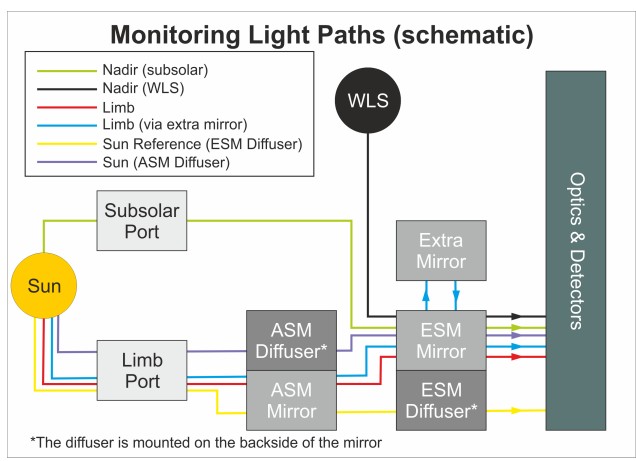

**Figure 1.** Schematic diagram of SCIAMACHY's scanner unit with all light paths relevant for the re-calibration of the solar spectra. The nadir light path provides solar measurements via the sub-solar port and ESM mirror, the limb light path solar measurements via the limb port and both scan mirrors. There is also the option to use an extra mirror in this light path. The nominal solar reference measurements (Sun reference, absolute radiometrically calibrated) uses the ESM diffuser and the ASM mirror as light path. Solar measurements via the ASM diffuser and the ESM mirror are also possible. In-flight monitoring and wavelength calibration are done using the WLS and the spectral line source (SLS, not shown), respectively.

## 3 Optimised Degradation Correction

### 3.1 Starting Point: In-flight Degradation Correction Approaches for SCIAMACHY

The general idea of the degradation correction for the SCIAMACHY instrument is to use a combination of measurements from different monitoring light paths to identify changes in the optical components (diffuser and mirror transmission changes).

The first version of a degradation correction was developed from the monitoring of the instrument and applied for SCIA-MACHY V7 level 1 products (solar irradiance and backscattered radiances) (e.g. Bramstedt et al., 2009). Ratios of solar and WLS monitoring measurement (see Figure 1) with respect to initial measurements from 2 August 2002 (beginning of SCIAMACHY measurements) were determined for the various optical paths. The time series of the ratios are the so-called monitoring factors (m-factors) and can be used for the degradation correction. This approach does not distinguish between

degradation and natural variability in solar radiation. Therefore any trend information in SCIAMACHY SSI is lost.

For SCIAMACHY V8 a more sophisticated approach was introduced and revised in V9. The scanner unit (mirrors and diffusers, see Figure 1) is described by a physical model (Krijger et al., 2014; Bramstedt, 2014). The model uses the Mueller matrix formalism and Fresnel equations to describe the different light paths and optical elements in the scanner unit (for details see Krijger et al., 2014). The approach follows the hypothesis that mirrors and diffusers degrade by deposition of a thin

absorbing layer of contaminants that changes with time. Thus, the thickness and optical properties of the contamination layer on the mirrors and diffusers are used as a fit parameter within the scanner model. The scanner unit is followed by the optical

bench module (OBM, including telescope, spectrometer, and detectors) that is common for all light paths. Consequently, the OBM transmission change is described by a single, wavelength-dependent degradation factor (OBM m-factor). Parameters (contamination layer thicknesses and properties, OBM m-factor) are fitted from ratios of in-flight calibration measurements (degradation modelling) using the combination of available light paths.

An initial approach to account for solar variability was introduced with V9. The degradation model includes now additional natural variability terms using solar proxies (Mg II index and F10.7 cm radio flux) as auxiliary fitting parameters. The impact of sunspot darkening was to a first order neglected.

The degradation correction accounts only for degradation changes, since the first measurements in space. For SCIAMACHY V9, the reference day of the degradation correction was set to 27 February 2003. All calibration change from pre-launch
until the reference date for the in-flight degradation correction are assumed to be accounted for by the on-ground to in-flight correction (Hilbig et al., 2018) that was introduced with V9.

The degradation correction works in general well, but there are some issues in the final phase of the mission (after 2009) and the early period in 2002. It provided a reasonably stable solar spectrum that is sufficient for most atmospheric applications, but requires modifications for solar applications, e.g. studies of solar variability.

## 3.2   Modifications for Solar Applications

The degradation correction applied in V9.01 was not sufficient for studies on SSI variability and trends due to the challenges encountered to separate adequately natural and instrumental changes. It also did not work well in the final phase of the mission after 2009 and the early period during 2002. In addition some short time seasonal patterns remained in the time series and artefacts were visible for the first few days following non-nominal decontaminations, e.g. in August 2003, December 2004,
and December 2008. For solar applications, modifications of the degradation model are needed.

The following issues arise in the current degradation correction. Firstly, for operational reasons, the sub-solar measurements (see Fig. 1) were not made in 2002. As a result the OBM m-factors before February 2003 were calculated assuming fixed settings for the contamination layers (as defined at the reference day in February 2003) and a constant Sun for the ESM diffuser light path. Additionally, temperature settings of the instrument were changed in February 2003. An additional solar
light path uses an extra mirror such that the light crosses the ESM mirror twice. The extra mirror is small and covers only a part of the solar disc in contrast to all other light path measurements. Overall, there are basically two types of direct solar observations made by SCIAMACHY. The first involves a diffuser, that scatters the solar light into a diffuse beam with some loss of intensity, the second uses a small aperture to reduce the amount of incoming light. The limb, sub-solar, and extra-mirror light paths (Fig. 1) belong to the latter type. The observations via a small aperture cause additional diffraction effects. Hence, for an
optimisation of the degradation correction, small aperture data (type 2) are now excluded from the new degradation modelling. Only solar measurements via a diffuser are selected. ESM diffuser solar measurements (nominal solar spectra) remain part of the model and solar measurements via ASM diffuser and ESM mirror are added here. This new degradation correction modifies the scanner or mirror model and includes an updated calibration of the scan angle dependence from in-flight measurements (see Section 3.3 below).

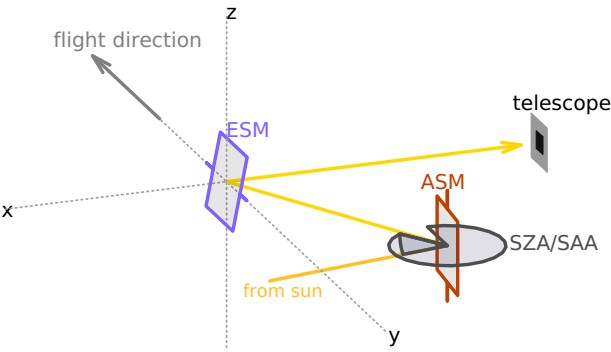

**Figure 2.** Sketch of SCIAMACHY's scanner unit with the light path for an ASM diffuser measurement. From the solar direction (defined by solar zenith and azimuth angles SZA/SAA), the light reaches first the ASM diffuser. A small fraction of the light is scattered towards the ESM mirror such, that the ESM mirror reflects the light into the telescope.

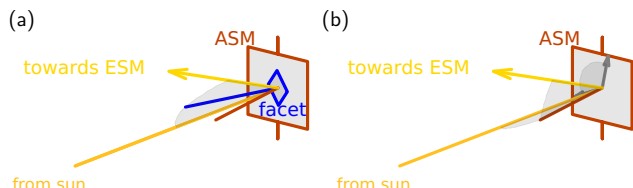

**Figure 3.** a) Only the ASM diffuser facets with an orientation for a specular reflection towards the ESM mirror contribute to the measured signal. The orientation of these facets are calculated from the direction towards the sun and the ESM mirror using analytic geometry. b) The diffuser sensitivity depends on the incident angles of the solar direction and the direction towards the ESM mirror, and the azimuth angle between these two.

In addition to the solar measurements, WLS measurements are included. The WLS is an independent light source, and therefore its measurements allow for a (better) separation of natural solar variability from instrument variations. To use these measurements in the degradation modelling, the change of the WLS with accumulated burning time needs to be properly characterised (see Section 3.4).

### 3.3 ASM diffuser solar measurements

During the on-ground calibration of SCIAMACHY, spectral features caused by the ESM diffuser were detected that were impeding atmospheric trace gas retrievals. In order to obtain a solar spectrum without these features it was necessary to add an additional diffuser with optimised optical properties on the backside of the ASM mirror assembly (Gottwald and Bovensmann, 2011). This enabled regular ASM-diffuser solar measurements to be made. Due to time constraints an absolute radiometric calibration for this second diffuser was not possible before launch.

The light path of the ASM diffuser measurements uses the limb port, the ASM diffuser and the ESM mirror (Fig. 1). The scanner model approach of the recent radiometric calibration for SCIAMACHY does not include yet the ASM-diffuser light path. The scanner model needs the incidence angles on the optical surfaces and the relative orientations along the light path. In SCIAMACHY the detector and both scanning mechanism are in one plane with the rotation axis perpendicular to each other (Fig. 2). This geometry simplifies the calculation of the involved angles (see details in Krijger et al., 2014). In the model, a diffuser is described as a roughened surface with tiny mirror facets. Only those facets with an orientation for a specular reflection into the instrument contribute to the signal. The polarisation behaviour of a diffuser is described by a Mueller matrix for a mirror with the orientation of the reflecting facets. This orientation and the rotation angle for the Stokes vector frame from the ASM diffuser facet to the ESM mirror is calculated numerically using analytic geometry. Input are the direction towards the Sun (defined by the solar zenith and azimuth angle) and the direction towards the ESM mirror (defined by the rotation angle of the ESM) as shown in Fig. 3a.

The diffuser sensitivity depends on the incidence angles of the in-coming irradiance, the out-going beam towards the telescope via the ESM mirror, and the azimuth angle between these directions (Figure 3b).

One ASM diffuser measurement comprises a sequence of 30 individual measurements. The ASM diffuser is rotated by 12° during the sequence to further minimise spectral features in the mean spectrum. Additionally, the observing geometry towards the Sun varies over the year, which leads to five different ranges for these rotations over the year. A closer look shows, that all possible geometries are covered in a time range from mid February to end of May. The ESM mirror stays at a fixed angle, therefore two of the three angles are sufficient to parameterise the diffuser sensitivity. We choose the incident angle from the solar direction and the azimuth angle. The degradation itself as well as the degradation rate is lowest in the early phase of the mission. Therefore, Sun-Earth distance normalised in-flight measurements from February to March 2003 are used to generate a wavelength dependent look-up-table (LUT) for the diffuser sensitivity.

Figure 4 visualises an example of the LUT. The LUT normalises each individual ASM diffuser measurement to the diffuser sensitivity of the reference measurement. The normalisation factors are in the range of $\pm 40\%$. After this normalisation, the differences between the individual readouts of a measurement sequence are well below 0.5 %.

On 26 May 2003, the tangent height for the dark current measurements of the limb measurement were changed by up-dating a so-called timeline of the instrument. This change had an unintended side-effect for the ASM diffuser measurement: The fixed position of the ESM mirror during the ASM diffuser measurement was changed by about 1°, which alters the range of possible geometries. Therefore, a second LUT has been generated from the measurements mid February to end of May 2004 to normalise the measurements since 26 May 2003.

With the integration into SCIAMACHY's scanner model and the calibration of the diffuser sensitivity from in-flight measurements, we provide a radiometric calibration for the ASM diffuser measurements relative to the reference measurement.

### 3.4 WLS ageing correction

SCIAMACHY's internal White Light Source (WLS) is a 5 W UV-optimised Tungsten Halogen lamp. Its primary role is to determine the pixel-to-pixel gain, check the overall throughput of the instrument, and correct wavelength dependent effects

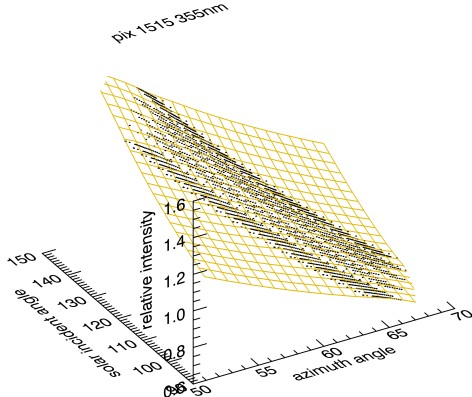

**Figure 4.** Visualisation of a look-up-table (LUT) for the ASM diffuser sensitivity. The LUT for 355 nm (in channel 2) is shown. The black dots are the measured intensities, relative to the reference measurements (15th readout on 27-Feb-2003). The overlay is the interpolated LUT. The LUT values depend on the incident angle of the solar irradiance and the azimuth angle between the solar and the ESM incidence angles.

(e.g. etalon effect, quantum efficiency). The WLS has been used already during the calibration activities on-ground and during the commissioning phase after launch. In nominal operation, the regular WLS measurements are performed weekly. In between, further WLS monitoring measurements using different optical elements in the light path are performed. Each nominal WLS measurement has a burning time of 12 s with the last 4 s used as measurement signal.

5    In the new degradation model, the WLS is used as an independent second light source. Before these measurements can be used in the degradation model, the "ageing" of the WLS itself has to be corrected so that the WLS can be considered as a constant light source.

Our assumptions on the WLS behaviour use the understanding and knowledge gained from a detailed study by Sperling et al. (1996) at the Physikalisch-Technische Bundesanstalt (PTB). Lamps, similar to the WLS used in SCIAMACHY, show 10    some typical ageing in the beginning until a more stable level is reached after some accumulation of burning time. Figure 5 shows as an example the change in WLS signal at 630 nm with accumulated burning time in space. The time series follows for most parts an exponential law (see the blue line in Figure 5). The WLS change with accumulated burning time $t_B$ can be parameterised with the following function

$$S_{WLS} = -p_0 \cdot exp[-p_1 \cdot t_B] + p_2, \tag{1}$$

15    where $S_{WLS}$ is the relative WLS signal with respect to the signal at the reference date of the degradation correction (27 February 2003). However, the time series indicate some deficits in the calibration for the early phase of the mission (2002) and the last years of operation (2009 – 2012). Therefore, only the red marked period of the time series is used in a non-linear least-square fit. The fit result (light blue curve in Fig. 5) gives the theoretically expected ageing of the WLS with time and is used

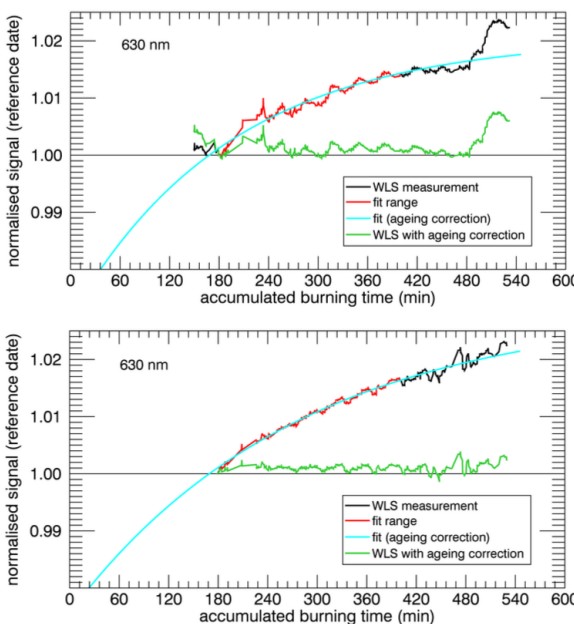

**Figure 5.** Relative WLS intensity at 630 nm as a function of its accumulated burning time in orbit. *Top:* WLS measurements in ESA V9.01 calibration; *bottom:* after the final iteration of the modified degradation model. The *black curve* shows the WLS measurements. The (fit) range that is used to derive the parameterization of the WLS change is indicated in *red*. The best fit to describe the ageing of the WLS is plotted in *light blue*. In *green*, the corrected WLS measurements as used in the degradation model is shown.

to correct the WLS signal accordingly. This ageing of the WLS is highly spectrally dependent and happens faster for shorter wavelengths.

This approach works well for the visible and NIR spectral range. Obtaining a similar parameterization of the WLS change with burning time in the UV (especially below 400 nm) turned out to be more problematic. The UV time series in SCIAMACHY

5   V9.01 are not sufficiently corrected for instrument degradation to define an adequate fit range. Therefore, it is not possible to obtain the parameterization of the WLS ageing directly by fitting an exponential law (Eq. 1). The parameters $p_0$, $p_1$, and $p_2$ received from fits in the visible and NIR were extrapolated to smaller wavelengths, which led to a first approximation for the WLS ageing parameterisation in the UV spectral range and enables to extend the WLS ageing correction down to 330 nm (SCIAMACHY Channel 2).

10    Remaining residual features in the ageing-corrected WLS time series show uncorrected degradation effects that originate from other optical elements in the light path. In order to account for them, an iterative approach was used by repeating the WLS ageing parameterization after the optimised overall degradation correction as described in the next section.

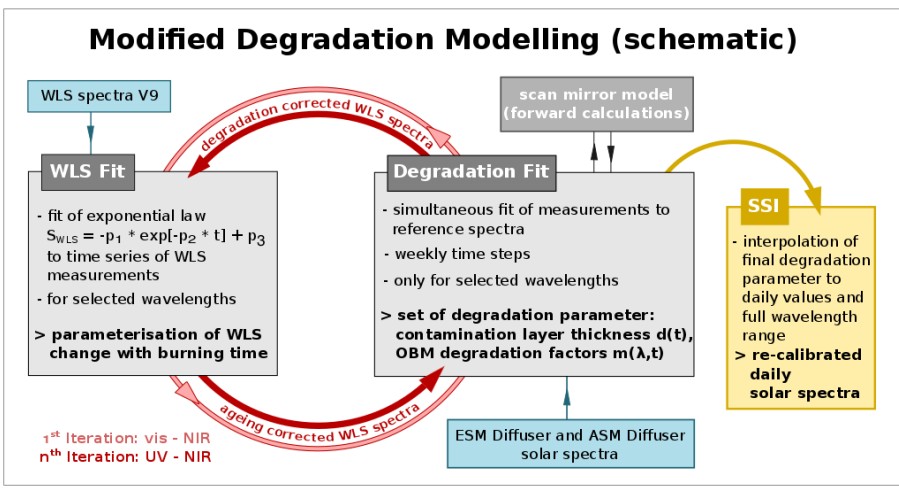

**Figure 6.** Schematic representation of the degradation modelling (Details see text).

## 3.5 Iterative degradation fit

A new set of parameters to describe the instrument degradation was derived by fits of in-flight measurements. In the optimised version of the degradation model solar measurements via ESM diffuser as well as measurements using the ASM diffuser (see Section 3.3) were included. Furthermore, the internal WLS represents a second independent light source. Its measurements
were corrected for lamp ageing beforehand (see Section 3.4).

A sketch of the modified degradation modelling is shown in Fig. 6. The fit of new degradation parameters (here contaminant layer thicknesses and OBM m-factors) is performed in weekly steps based on the sampling of the WLS measurements and only for selected wavelengths. The scanner unit model (Krijger et al., 2014) is used in the forward calculations. All three measurements (ESM diffuser, ASM diffuser, WLS) are fitted simultaneously. The derived parameters are time series of the
contamination layer thicknesses of ESM mirror, ASM mirror, and ASM diffuser as well as the OBM degradation factors (summarising all spectrometer transmission and detector changes). An initial guess and first approximation of these parameters are the m-factors derived from the ESA V9.01 calibration. Following the experience from the V9.01 correction, the contaminant layer on the ESM diffuser is considered as constant and the small ESA V9.01 value from the reference date (27 February 2003) is used for all time steps.

The results of the first degradation fit are used to repeat the WLS fit (see Section 3.4). With the second set of ageing corrected WLS spectra, a second degradation correction is generated and so on. Already after the first iteration of the new degradation modelling the results are quite close to the final ones. After three iterations of WLS fitting and subsequent degradation modelling, the final results are obtained.

The lower panel of Fig. 5 shows the performance of the WLS ageing correction after the final iteration. The seasonal
variations are much smaller and the WLS measurements later than 2008 now clearly follow the expected curve. Figure 7

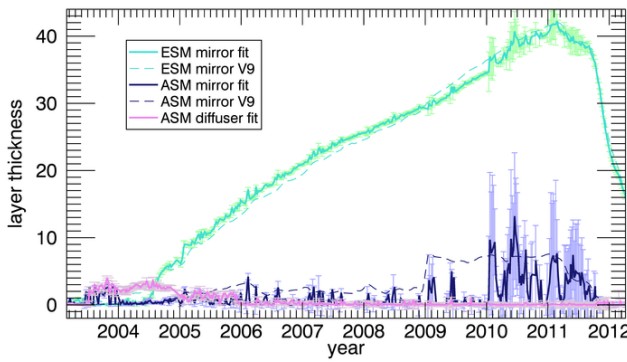

**Figure 7.** Contamination layer thicknesses of the ESM mirror, ASM mirror, and ASM diffuser as a function of time (mission lifetime in orbit). For each optical element, two data sets are shown: dashed lines indicate the layer thicknesses of the ESA V9.01 degradation correction and bold lines (including error bars) show the layer thicknesses derived with the modified degradation model from this study.

shows the variation of the contamination layer thicknesses for the relevant elements of the scanner unit (see Fig. 1) over the SCIAMACHY mission lifetime. The OBM degradation factor and its changes are illustrated in Figure 8 by some selected wavelengths from UV to NIR. The OBM is mostly degrading in the UV (SCIAMACHY spectral channels 1 and 2) and the NIR (Channel 6). One prominent jump in the time series at the beginning of 2009 has its origin in an ice decontamination

period in December 2008. In addition, the corresponding values of SCIAMACHY V9.01 are shown as dashed or thin lines in the background of each graph. The uncertainties of the individual degradation parameter are derived within the fit procedure and indicated in the figures. Large fluctuations as well as highest uncertainties of the degradation parameter arise in the last years starting with 2010. Therefore, comparisons of the resulting SCIAMACHY solar spectral irradiance time series with other measurements and model reconstructions include only data before 2010.

The final set of degradation parameter is interpolated to daily values and in case of the wavelength dependent OBM degradation factors to the SCIAMACHY spectral scale (identical with V9.01) between 320 to 1600 nm. Due to the Wood anomaly feature around 350 nm (Liebing et al., 2018), the spectral range 340 – 360 nm was omitted in the fit and only two wavelengths, 330 nm and 370 nm, for SCIAMACHY's spectral channel 2 (300 – 400 nm) were included. The total available wavelength range was defined firstly by the limit in deriving the WLS ageing correction which was only possible starting from 330 nm.

Secondly, wavelengths above 1600 nm (SCIAMACHY channel 6+, 7, and 8) were not used in the optimised degradation modelling. Remaining uncorrected short time seasonal patterns and instrument degradation increase above 1600 nm. Since the impact of changing contamination layer thicknesses is low in this wavelength range, the inclusion in the degradation modelling could induce artefacts in the resulting degradation parameter. The newly derived parameters are then applied to re-calibrate the SCIAMACHY SSI measurements.

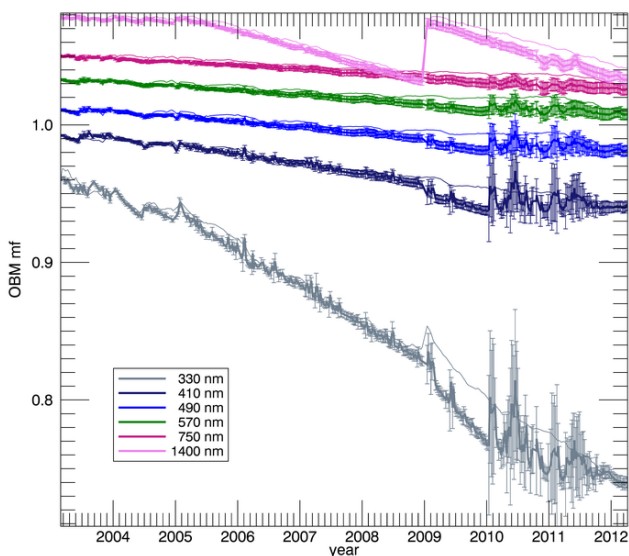

**Figure 8.** OBM m-factors for some wavelengths as a function of time (mission lifetime in orbit). For each wavelength, two data sets are shown: thin lines indicate values from the ESA V9.01 degradation correction and thick lines with error bars show the results from the modified degradation model from this study. The time series are vertically shifted for better visibility.

## 4 The recalibrated SCIAMACHY SSI

The optimised degradation correction is used to re-calibrate the SCIAMACHY solar spectral irradiances in the wavelength range 320 – 1600 nm. Figure 9 shows time series of SCIAMACHY solar spectra at selected wavelengths. The solid lines are re-calibrated SSI data from this work, and for comparison the solar spectra of SCIAMACHY ESA V9.01 are drawn as
dashed lines. With respect to ESA V9.01, a general improvement can be seen in the time series, especially for measurements after 2008. The new results show a more stable signal in the NIR with less small-scale structure. Reasonable results are also obtained in the UV near 330 nm. Here ESA V9.01 shows variations that can not be attributed to natural solar variability, while our results show more clearly the continuous decrease as expected from the descending phase of solar cycle 23 with a clear minimum at 2008/2009 as evident from other SSI measurements (e.g. Mauceri et al., 2018). However, the values are higher than
expected from TSI variability ($\sim 0.1\,\%$ solar cycle; Kopp, 2016). Furthermore, an unexpected anti-cyclic increase during solar minimum, similar to the behaviour of V9.01, becomes evident in the NUV and above; see further discussion and comparisons with other SSI data sets in Section 5.

   Some new artefacts are visible, e.g. at the beginning of 2011. They seem to originate in the ASM diffuser solar spectra and are transferred to the ESM diffuser solar spectra and WLS spectra through their simultaneous fit in the degradation modelling.
Besides, a beginning recovery of the instrument throughput was recorded which is not fully understood. A possible explanation is the additional permanent use of a backup system after an anomaly in the Ka-band antenna subsystem. About 120 W more energy were dissipated resulting in a change of the thermal environment of Envisat. Increased temperatures (0.3 – 2.7 °C,

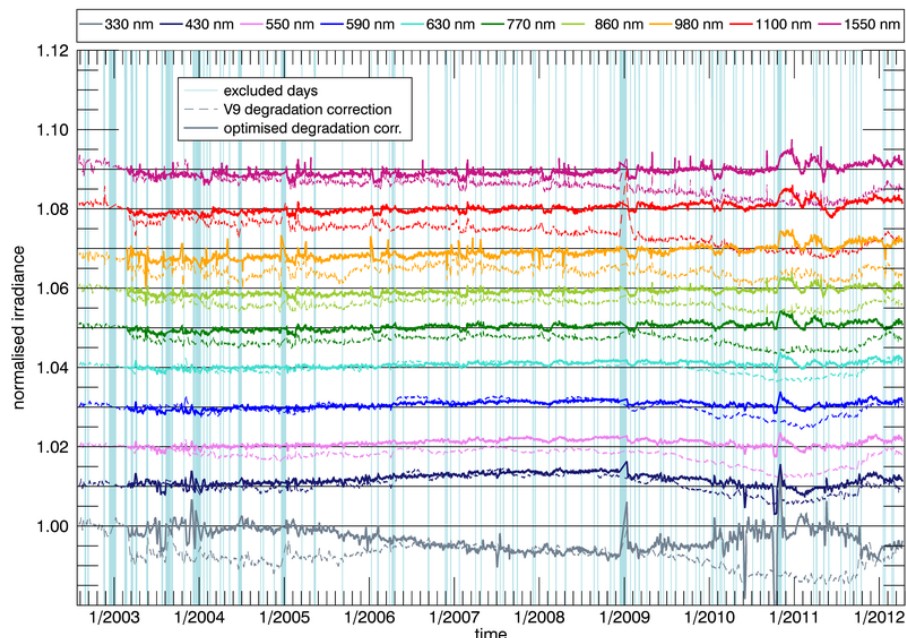

**Figure 9.** SCIAMACHY SSI time series for selected wavelengths. Each time series is shifted vertically by 0.01 for better visibility and normalised to the reference day of degradation correction (27 February 2003). The dashed lines show the SCIAMACHY SSI with ESA V9.01 calibration; solid lines the re-calibrated SCIAMACHY SSI from this work. Vertical bars indicate days or extended periods with maintenance activities (e.g. ice decontamination periods), platform and instrument anomalies, and orbit control manoeuvres when data are considered highly uncertain.

subsystem dependent) were observed by SCIAMACHY's internal temperature sensors (Gottwald et al., 2016). This changes the outgassing and thus the photochemical processing. At about the same time, the orbit of Envisat was lowered in October 2010. As a consequence, we attribute the higher variability and larger uncertainties in the degradation parameters during the final phase of the mission to the combination of these effects and advanced ageing of the instrument. Therefore, it is challenging 5 to generate a reasonable degradation correction for the last phase of the mission.

## 5  Comparison with SCIAMACHY SSI

The recalibrated SCIAMACHY SSI were compared to other measurements from satellites that have been briefly described in the introduction. In addition, our data were also compared with semi-empirical SSI reconstructions that are currently favoured in climate model simulations (Matthes et al., 2017). The SSI reconstructions are briefly described in the following before the 10 results of the SSI comparisons are presented and discussed.

## 5.1 SSI reconstructions

An alternative to long-term SSI observations are estimates of long-term solar variations from measurements of shorter periods (e.g. a few solar rotations) by establishing information about long-time variations from extrapolation in time of fitted solar proxies, like sunspot number, Mg II index or 10.7 cm radio flux. Wavelength-dependent scaling factors of the solar proxies

can then be used to determine SSI changes (e.g. Lean et al., 1997; Morrill et al., 2011; Pagaran et al., 2009, 2011). The Mg II index is strongly correlated with SSI changes of the near and extreme UV on timescales of the 27-day solar rotation and the 11-year solar cycle. It is thus a common solar proxy for UV SSI variability (e.g. Viereck et al., 2001; Dudok de Wit et al., 2009; Snow et al., 2014) and often used to describe solar forcing in the upper atmosphere. Another key indicator of solar magnetic activity is the total area of sunspots. The radiometric effect of sunspot passages across the solar disc can

be quantified by the photometric sunspot index (PSI) (Balmaceda et al., 2009). A more complex approach by combining results from solar proxy models and solar atmosphere modelling is employed in the Naval Research Laboratory's (NRL) solar irradiance variability model, namely NRLSSI2 (Coddington et al., 2016). Another class of models uses semi-empirical models of the solar atmosphere to calculate the brightness of different surface features (such as sunspots, faculae, plage). An example is SATIRE-S (*Spectral And Total Irradiance REconstructions for the satellite era*) that derive the surface area coverage of the

individual photospheric components by full-disc intensity images and magnetograms (Yeo et al., 2014).

## 5.2 SSI time series

For comparisons with the re-calibrated SCIAMACHY SSI, we use the most recent data versions of SATIRE-S[2], SIM[3], and OMI[4]. SIM obtains daily solar spectra from 240 to 2400 nm at a variable spectral resolution of 1 to 34 nm with an absolute uncertainty of 2 %, and long-term repeatability of less than 0.1 % (Harder, 2019). Before any comparisons were performed, as

shown in Figures 10 and 11, outliers are removed from SIM V25 time series. OMI provides daily solar spectra in the 265 – 500 nm range of mid resolution (0.4 – 0.6 nm) with an absolute accuracy of better than 4 % and high instrument stability. The optical degradation rate ranged from 0.2 to 0.5 % yr$^{-1}$ (Marchenko et al., 2016, and references therein). The authors note that the current values underestimate the solar cycle variability by $\sim 0.1$ % in the UV ($< 350$ nm) and $< 0.05$ % in the visible, which originates in the applied degradation correction approach.

Figure 10 shows SSI time series for several wavelength bands. Measurements from different instruments and model reconstructions are compared with SCIAMACHY SSI. The data are normalised to a reference date during solar minimum condition (October 5, 2008). The solar cycle minimum occurred by the end of 2008 / beginning of 2009. We choose this specific date as it lies within a period of stable SCIAMACHY measurements where uncertainties are smaller than in early 2009. The panels show the data starting with the reference date of SCIAMACHY degradation correction (February 27, 2003) until the end of

2009. As discussed in Section 4 and shown in Figs. 7 and 8 the degradation modelling had larger uncertainties after 2009 when

---

[2]SATIRE-S: https://doi.org/10.17617/1.5U, Accessed: 17 Sep. 2018
[3]SIM V25 (Harder, 2019)
[4]OMI: https://gs614-sbuv-pz.gsfc.nasa.gov/solar/omi/lisird/readme_omi_irradiance.txt

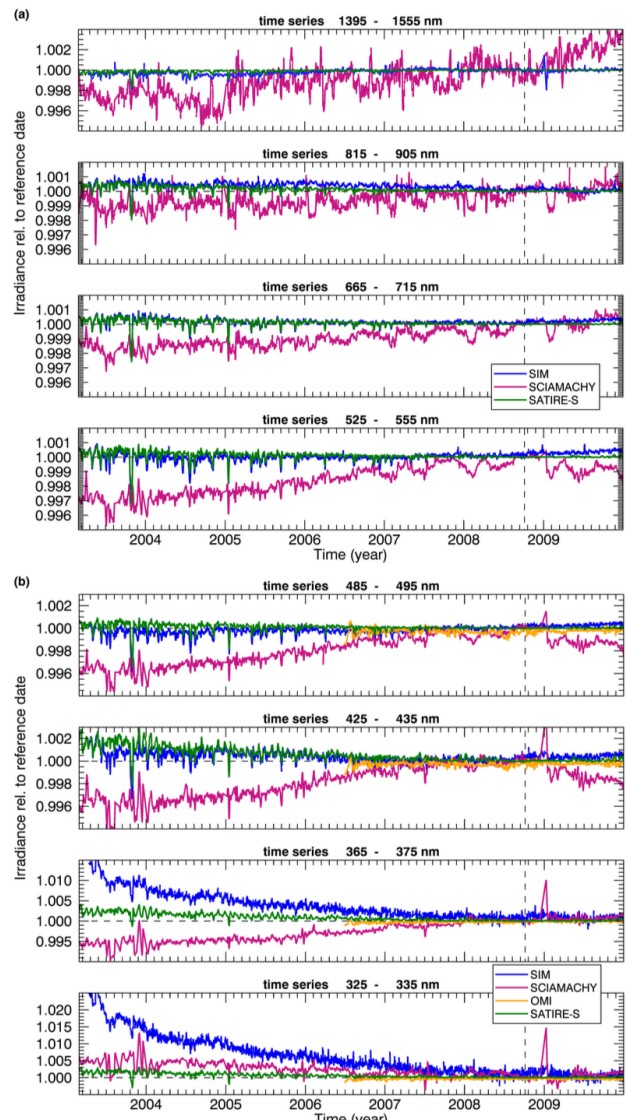

**Figure 10.** Time series of SSI, integrated in (b) 10 nm and (a) 30 – 160 nm wide wavelength bands from SCIAMACHY, SIM/SORCE V25, OMI/AURA satellite measurements and SATIRE-S. The time series are ratioed to values from October 5, 2008, which are representative for solar minimum conditions. Details see text. (Note the change in vertical scale for the different wavelength ranges.)

instrument degradation became more prominent. For our comparisons between different data sets, we focus in the following on the descending phase of solar cycle 23 until the end of 2009.

Beginning with the lowest wavelength band centred at 330 nm, the SCIAMACHY SSI time series lies within the solar cycle variation of SATIRE-S and SIM V25 data sets. It shows the expected minimum at time of solar activity minimum in accordance with the other SSI data. Nevertheless large differences in comparison to SIM become obvious in the UV. This

was previously reported by studies from e.g. Haberreiter et al. (2017); Mauceri et al. (2018); Coddington et al. (2019) that point to comparably large solar cycle variability results for recent SSI from SIM in the UV. For higher wavelengths in the UV and visible spectral range (starting at 370 nm in Fig. 10) SCIAMACHY shows an increasing signal towards solar minimum and decrease afterwards. This would imply anti-correlation of the solar irradiances in the visible and the 11-year solar cycle.

Similar results were published earlier for SIM (Harder et al., 2009; Haigh et al., 2010) but are inconsistent with other satellite measurements that show in-phase variations (Wehrli et al., 2013; Marchenko and DeLand, 2014). Recent studies by Woods et al. (2018) and Mauceri et al. (2018) developed new methods to account for uncorrected degradation in SIM SSI. Both results, the MuSIL-corrected SIM and SIMc, show better agreement with independent SSI data such as SATIRE-S than the operational SIM product (Harder, 2019). The observed anti-correlation for the new SCIAMACHY results is therefore likely a

remaining residual instrument artefact. For the time series in the NIR above 1000 nm (SCIAMACHY Channel 6) the overall increase agrees qualitatively with the SIM observations, but the rate of increase is higher than compared to SIM and SATIRE-S. OMI generally shows good agreement with SATIRE-S and therefore, strengthen its role as reference for SSI comparisons. Unfortunately, the degradation corrected data set starts in July 2006 and is not available for the first part of the mission when solar activity was stronger.

It seems that for most parts of the visible and near IR spectral range a remaining positive drift, most likely instrumental in nature, is evident in the SCIAMACHY time series that is on the order of +1%/decade. The SCIAMACHY mission was accompanied by a high number of instrument and platform anomalies as well as many regular maintenance activities as illustrated by blue lines in Figure 9. These anomalies and maintenances had mostly minimal impact on atmospheric trace gas retrievals, the primary purpose of SCIAMACHY, but has non-negligible impact on the degradation correction and radiometric stability.

One of the most important maintenance activities were decontamination periods where the detectors were heated to remove ice contamination of the NIR detectors, but they also impacted the UV and visible spectral bands.

In 2004, larger short time variations are visible in the SCIAMACHY time series. At the beginning of the mission the contamination layers are still faint and the resulting degradation effect small. Therefore it is more difficult to determine the degradation parameter in the degradation modelling. That is further hindered by ice decontamination periods that were included

in the instrument operation more frequently at the beginning of the mission.

On smaller time scales (solar rotations), SCIAMACHY follows most of the prominent signatures in the time series, see Figure 11. Up to about 700 nm SCIAMACHY shows realistic decrease in the amplitude of variations from higher solar activity towards the minimum.

Despite a clear improvement of the SCIAMACHY SSI in comparison with ESA V9.01, as discussed in Section 4, the current

degradation correction is not yet sufficient to account for all instrumental effects, such as the remaining positive drift in the SSI time series. Further work is required to clarify the origins of differences between SCIAMACHY and other SSI data sets. Nevertheless, we were able to show that the white-light-source (WLS) in the new degradation correction scheme significantly contributes to improvements in the in-flight radiometric calibration.

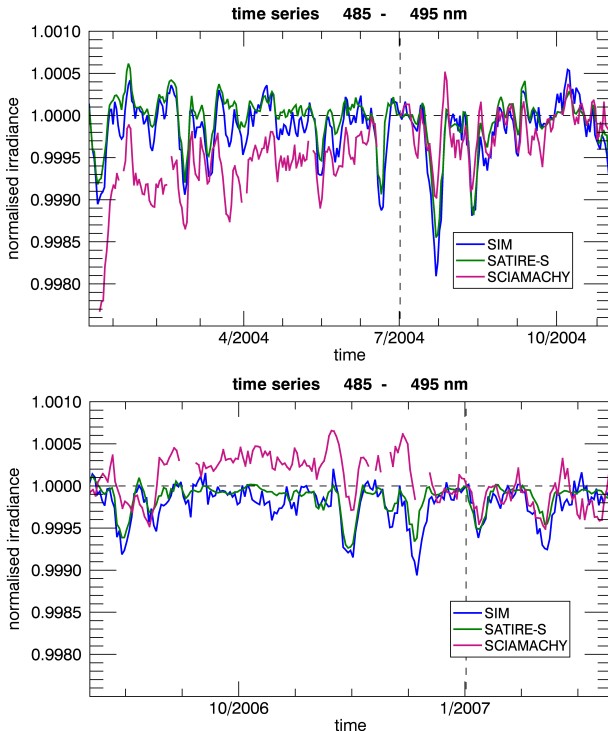

**Figure 11.** Time series of SSI, integrated in 10 nm wide wavelength band from SCIAMACHY, SIM/SORCE V25, and SATIRE-S. The Figure provide a detailed view of 10-months periods in 2004 (top) and 2006/2007 (bottom). The time series are ratioed to values from July 1, 2004, and January 1, 2007, respectively, to shift the data sets for better comparability.

## 6 Summary and conclusions

One of the main limitations for long-term space-based measurements is the optical degradation of the instrument due to the harsh environment of space operation. Therefore, a thorough degradation correction to maintain instrument calibration for the instrument's lifetime is required. In this paper, we presented a modification of SCIAMACHY's degradation correction approach (Krijger et al., 2014), which models (growing) contamination layers on the optical surfaces in the scanner unit of the spectrometer being responsible for instrumental throughput losses with space mission time. Changes in the optical components (diffuser and mirror transmission changes) were identified using a combination of solar measurements from different monitoring light paths. In this work the set of solar monitoring measurements to be included in the degradation modelling was changed. Solar measurements via a second diffuser (ASM diffuser) are now included in the model, after new scan angle dependent transmission changes were introduced. The second essential part of the modification is the inclusion of measurements from SCIAMACHY's internal white light source. Including the WLS as a second and independent light source provided the opportunity to better separate instrument variations and natural solar variability.

Before the WLS measurements could be used in the degradation model, emission changes of the WLS had to be accounted for. The change of the WLS emission as a function of accumulated burning time was successfully determined, which was found to be qualitatively consistent with detailed lamp studies by Sperling et al. (1996) at the Physikalisch-Technische Bundesanstalt (PTB). An iterative approach of WLS correction and subsequent spectrometer degradation modelling was successfully

implemented.

A general improvement over the current SCIAMACHY ESA V9.01 SSI was found, but there are still limitations in the degradation corrections, particularly in the late period between 2010 until the end of the satellite mission in 2012. Due to the numerous instrument and platform anomalies as well as intended decontamination phases to sublime the ice layer on the SCIAMACHY detectors, the degradation corrections were not sufficient to remove all artefacts and small drifts in the

SCIAMACHY SSI time series.

The main achievement of this study was the successful characterisation of the WLS ageing and the integration of the WLS measurements in the existing degradation model. This will provide an important contribution for a revised degradation correction in possible future data products by ESA. This study showed the potential of an internal WLS for degradation monitoring of other satellite instruments such as GOME-2, OMI and TROPOMI, that also include this type of lamps.

*Data availability.* The newly derived SCIAMACHY SSI data set will be available at http://www.iup.uni-bremen.de/UVSAT/datasets/scia_ssi_timeseries. Please note that there are still limitations in the degradation correction and not all calibration issues were solved. At the time of writing, the SCIAMACHY SSI data set covers the wavelength range from 320 to 1600 nm and contains gaps at SCIAMACHY spectral channel boundaries and around 350 nm. SCIAMACHY provided daily measurements with short interruptions during satellite/instrument maintenance periods. Possible updates will be provided/announced on the web page.

*Acknowledgements.* SCIAMACHY is a national contribution to the ESA Envisat project, funded by Germany, The Netherlands, and Belgium. The authors thank ESA for providing the SCIAMACHY data and the SCIAMACHY Quality Working Group for their work on SCIAMACHY level 1 data. The support from the SCIASOL project under the BMBF priority programme ROMIC (Role of the Middle Atmosphere in Climate) and the University of Bremen is gratefully acknowledged.

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
