# Peer review of "Optimised degradation correction for SCIAMACHY satellite solar measurements from 330 to 1600 nm by using its internal white light source"

_Atmospheric Measurement Techniques, 2019_

## Referee Comment (RC1) · Anonymous Referee #1 · 6 Jan 2020

GENERAL SUMMARY AND COMMENTS

This paper presents a revised analysis of the long-term degradation of the SCIA-MACHY instrument flown on the Envisat satellite, focusing on corrections for solar spectral irradiance (SSI) measurements. Since SCIAMACHY did not make end-to-end calibration measurements on-orbit, data from an internal white light source (WLS) are used in combination with a physical model of optical surface contamination to characterize instrument changes.

This paper is well-written, with good discussion of the procedures that were developed and the key results. Some suggestions and comments related to specific items are

provided below.

SPECIFIC COMMENTS

1. p. 4, line 3: As a point of terminology, I would call the results presented in Hilbig et al. (2018) the "absolute radiometric calibration", whereas the work presented here improves the relative (or time-dependent) instrument calibration.

2. p. 9, lines 2-3: I'm not sure about this assumption. Many instruments experience the most rapid degradation early in their lifetime, when contaminants are fresh. It is true that cumulative degradation in early 2003 will be small compared to the end of the SCIAMACHY mission.

3. p. 10, lines 11-13: What is the typical amount of burning time per year for the WLS? Figure 5 suggests ∼40-50 minutes per year during 2003-2012, although apparently there was more usage during 2002 (∼180 minutes?) that is not used for the degradation correction. Does the statement on p. 11, lines 18-20 mean that the WLS was used weekly? If so, this would imply only ∼1 minute of operation during each sequence, which represents a fairly short duration for an on-orbit lamp to reach stable operating conditions. Please comment.

4. p. 11, lines 4-6: How does the uncertainty in the degradation fit change with wave-length? The relative uncertainty at 630 nm shown in Figure 5 [bottom] is clearly small. Was there a specific threshold that caused you to set 330 nm as a lower limit? I ask in part because corrected SCIAMACHY SSI data covering 212-330 nm would be a valuable addition to the SSI database during 2003-2012.

5. p. 11, line 18: Figure 6 is very useful.

6. p. 12, lines 12-14: I don't understand why the ESM mirror appears to have a steady buildup of contaminant throughout the mission (Figure 7), while the ESM diffuser has a constant layer of contaminant (p. 12, lines 4-5). Is there a large difference in exposure to contaminating material between these two elements?

7. p. 13, lines 3-5: The magnitude of the SCIAMACHY solar cycle decrease at 330 nm (∼0.8%) is still somewhat larger than would be expected from TSI change (∼0.1-0.2%). Meanwhile, the 430 nm time series shows an increase of ∼0.3-0.4% that is out of phase with the TSI variation. This result is discussed more extensively in Section 5.2, but it might be helpful to mention it here.

8. p. 17, lines 1-2: I feel that the authors have done a lot of excellent work to reach this level of accuracy.

9. p. 17, lines 12-14: The SCIAMACHY SSI data are valuable for studies of short-term solar variations because they have sufficient spectral resolution in the visible and near-IR to provide unique information about the behavior of solar absorption features in those spectral regions.

10. p. 18, lines 24-25: Are there any plans to release these revised SSI data? Is there a timetable for the creation of an ESA Version 10 data product?

TYPOGRAPHICAL ERRORS

p. 1, line 22: "simultaneous" should be "simultaneously".

p. 6, line 9: "begin" should be "beginning".

p. 6, line 31: "reasonable" should be "reasonably".

p. 12, line 12: "now clearly the" could be changed to "now clearly follow the".

---

## Referee Comment (RC2) · Anonymous Referee #2 · 13 Jan 2020

General comments:

This paper deals with new progress about the instrumental corrections of the SCIA-MACHY data. The special interest is to improve the solar spectral data. The paper is well written and contains interesting and important results for users of SCIAMACHY data. I would like to recommend the publication of this paper in AMT. I have the following minor comments and questions.

Specific comments:

1. p2, line 27: "Shortly after SCIAMACHY,…". Add "launch" after SCIAMACHY.

[Figure]

2. p3, line15: Please add some comments about the contamination problem: Is SCIA-MACHY's contamination problem worse or better when compared with similar instruments? If worse, what went wrong?

3. p3, line 15: Please, tell briefly about water and the other contaminants: Are they always on separated surfaces (mirrors, diffusers, detectors)?

4. p3: Perhaps you could say something about occultation instruments, which are to some extent more resilient to contamination problems due to the self-calibrating property.

5. p7, line 10: Another kind of. . . . Is this story part of the story on line 13 where you tell that "there are basically two types. . ."? I am confused.

6. p7, line 25: You have detected spectral features by the ESM diffuser. On line 26 you say that's why the ASM diffuser was added but without pre-launch calibration. Are you sure that ASM does not have similar spectral features than ESM?

7. p12, line 5: I get the impression that the ASM diffuser contamination increases along the mission, but the ESM contamination remains constant. Is this right and what is the reason?

8. p14, line 5: Could you tell something more about the change of the thermal environment?

9. p15, line 22: outlier ->outliers

10. p16, line 5: At 330nm in Fig. 10 the SIM curve seems to deviate clearly from the other curves, but your comment is more positive. Could you explain?

11. p18, Conclusions: Your conclusions are quite upbeat which is understandable after very tedious and extensive work. But if I consider the results shown in Fig.10 to be the most important outcome, the situation is not looking very promising. Is it possible that something important is still waiting to be found?

---

## Referee Comment (RC3) · Martin Snow (Referee) · 3 Feb 2020

In general, this is a good paper and I recommend only a few minor corrections.

Section 3.4 is about the aging correction for the WLS. As I understand your text, the WLS is only used to make a flat-field correction to the detector array. Corrections to the long-term trends in the SSI time series come from the other sources. Is this correct?

In the new degradation model, the WLS is an "independent" light source. The corrections shown in Figure 5 and described in Equation 1 seem to assume that all of the observed changes in the lamp are due to degradation of the lamp and none of the loss

of signal is due to the rest of the system. Fitting a curve to remove the variation in the WLS seems to make an assumption about the magnitude of changes in the detector (for example). If I have interpreted the text correctly, can you add some discussion on how this changes the trends in the SSI time series? If I have not interpreted the text correctly, then could you add another paragraph explaining how the WLS correction does not impact the final degradation correction?

In Figure 9, the 430 nm time series shows out of phase trends. Is this a statistically significant result about the Sun, or does this fall within the uncertainty of your SSI time series? In Figure 10, the 425-435 band also shows this behavior. In the text, you do mention the out of phase behavior in the SCIAMACHY data, but you don't make a clear statement on whether or not this is a new finding of SCIAMACHY. I would like to see a clarification on this point.

P16 L12: you refer to Woods' MuSIL and Mauceri's SIMc as if they were observational datasets. Both are essentially corrections based on proxies (or TSI) rather than instrument data. So comparisons to those time series should fall in the same category as comparisons to SATIRE-S or NRLSSI2.

Figure 11 uses different colors for the different missions than Figure 9. I would recommend that each instrument in the two figures have a consistent color assigned. It will make it easier for the reader to compare the time series.

---

## Author Comment (AC1) · 24 Apr 2020

We thank the reviewer for the detailed remarks and the efforts to improve our paper. We responded to all comments as best as we can. We added Section 5.2 of the manuscript as supplement.

**GENERAL SUMMARY AND COMMENTS**
This paper presents a revised analysis of the long-term degradation of the SCIA-MACHY instrument flown on the Envisat satellite, focusing on corrections for solar spectral irradiance (SSI) measurements. Since SCIAMACHY did not make end-to-end

calibration measurements on-orbit, data from an internal white light source (WLS) are used in combination with a physical model of optical surface contamination to characterize instrument changes. This paper is well-written, with good discussion of the procedures that were developed and the key results. Some suggestions and comments related to specific items are provided below.

**SPECIFIC COMMENTS**

**1. p. 4, line 3:** As a point of terminology, I would call the results presented in Hilbig et al. (2018) the "absolute radiometric calibration", whereas the work presented here improves the relative (or time-dependent) instrument calibration.

To clarify this point we suggest the following change:

*"The goal of our investigations is to improve the absolute radiometric calibration of the solar extraterrestrial spectrum measured by SCIAMACHY. This is achieved by adaption of the degradation correction for the SCIAMACHY instrument by..."*

is changed to:

*"The goal of our investigations is to improve the absolute radiometric calibration of the time series of the solar extraterrestrial spectra measured by SCIAMACHY. The degradation is derived relative to a reference measurement with published absolute calibration (Hilbig et al. 2018). ..."*

**2. p. 9, lines 2-3:** I'm not sure about this assumption. Many instruments experience the most rapid degradation early in their lifetime, when contaminants are fresh. It is true that cumulative degradation in early 2003 will be small compared to the end of the SCIAMACHY mission.

SCIAMACHY does not show a rapid degradation immediately after launch. Degradation occurs in the early phase of the mission, but the degradation rate increases over time until about 2006. The degradation rate is then stable until 2011. Thereafter a rapid recovery of the throughput started. We attribute the recovery to a changed thermal environment of the platform and thus of the atmosphere around the platform (see also issue 6 below). A change of outgassing rates from the platform leads to time-varying growth of an absorbing layer on the mirror.

To clarify this point we suggest the following change:

*"Assuming that in the early phase of the mission the degradation of the instrument is small, Sun-Earth distance normalised in-flight measurements from February to March 2003 are used to generate a wavelength dependent look-up-table (LUT) for the diffuser sensitivity."*

is changed to:

*"The degradation itself as well as the degradation rate is lowest in the early phase of the mission. Therefore, Sun-Earth distance normalised in-flight measurements from February to March 2003 are used to generate a wavelength dependent look-up-table (LUT) for the diffuser sensitivity."*

**3. p. 10, lines 11-13:** What is the typical amount of burning time per year for the WLS? Figure 5 suggests 40-50 minutes per year during 2003-2012, although apparently there was more usage during 2002 (180 minutes?) that is not used for the degradation correction. Does the statement on p. 11, lines 18-20 mean that the WLS was used weekly? If so, this would imply only 1 minute of operation during each sequence, which represents a fairly short duration for an on-orbit lamp to reach stable operating conditions. Please comment.

WLS measurements before 2nd August 2002 (start of regular operations) are special measurements during the calibration activities on-ground and during the commissioning phase just after launch and before nominal operation. Figure 5, top panel, shows all regular WLS measurements, meaning WLS over ESM mirror as shown in Fig. 1. More precisely, each WLS measurement has a duration of 12s. The first 8s (4 detector readouts) are omitted, the last 4 s (2 detector readouts) have a stable signal and are used.

These regular WLS measurements are performed weekly. In between, further WLS monitoring measurements using different optical elements in the light path are performed. Consequently the latter are not used for deriving the WLS ageing correction (and are not shown in Fig.5). The burning time is accounted for in the accumulated burning time. The WLS is a life-limited item of the instrument with a total budget of 25 hours. About one-third of this budget was used during the mission.

We added more details in the text, Sec.3.4 (bold).

*"SCIAMACHY's internal White Light Source (WLS) is a 5 W UV-optimised Tungsten Halogen lamp. Its primary role is to determine the pixel-to-pixel gain, check the overall throughput of the instrument, and correct wavelength dependent effects (e.g. etalon effect, quantum efficiency).* **The WLS has been used already during the calibration activities on-ground and during the commissioning phase after launch. In nominal operation, the regular WLS measurements are performed weekly. In between, further WLS monitoring measurements using different optical elements in the light path are performed. Each nominal WLS measurement has a burning time of 12s with the last 4s used as measurement signal.** *In the new degradation model, the WLS is used as an independent second light source. . . ."*
**4. p. 11, lines 4-6:** How does the uncertainty in the degradation fit change with wavelength? The relative uncertainty at 630 nm shown in Figure 5 [bottom] is clearly small. Was there a specific threshold that caused you to set 330 nm as a lower limit? I ask in part because corrected SCIAMACHY SSI data covering 212-330 nm would be a valuable addition to the SSI database during 2003-2012.

For smaller wavelengths the uncorrected instrument degradation (not WLS itself) becomes more prominent already after the first years/earlier in the mission. Therefore, it is more problematic to define an adequate fitting range to derive the WLS ageing. The fit residuals of the WLS ageing correction increase with decreasing wavelengths.

On the one hand the instrument degradation is strongest at smallest wavelengths, therefore these wavelengths would be beneficial in the degradation modelling. On the other hand the WLS ageing correction for wavelengths below 400 nm cannot be derived at the same quality as for the visible (as shown in Fig. 5). Due to the Wood anomaly feature around 350 nm (Liebing et al., 2018), we avoided the spectral range 340 – 360 nm. We therefore decided to only include two wavelengths for SCIAMACHY's spectral channel 2 (300 – 400 nm) in the fit.

In the paper, we added the following (Sec. 3.5, p. 13): *"Due to the Wood anomaly feature around 350 nm (Liebing et al., 2018), the spectral range 340 – 360 nm was omitted in the fit and only two wavelengths, 330 nm and 370 nm, for SCIAMACHY's spectral channel 2 (300 – 400 nm) were included."*

We fully agree, that an extension to lower wavelengths would be highly beneficial. With further improvements of the approach it might be possible to include wavelengths down to about 300 nm, but the correction of SCIAMACHY's spectral channel 1 (212 – 300 nm) remains challenging.

**5. p. 11, line 18:** Figure 6 is very useful.

**6. p. 12, lines 12-14:** I don't understand why the ESM mirror appears to have a steady buildup of contaminant throughout the mission (Figure 7), while the ESM diffuser has a constant layer of contaminant (p. 12, lines 4-5). Is there a large difference in exposure to contaminating material between these two elements?

We assume, as supported by earlier studies of SCIAMACHY optical degradation, that the accumulated exposure to solar UV radiation is closely related to the degradation of the optical elements of SCIAMACHY. The observed behaviour is consistent with the hard UV photolysing outgassed molecules (e.g. organic compounds from the printed circuit boards and water vapour carried on surface of the platform) in a thin atmosphere around the ENVISAT. The photochemical processing initiated by the UV creates low volatility species, which deposit. They absorb much more strongly in the UV than in the visible.

There is a large difference in exposure time to UV radiation for the ESM mirror and diffuser. The ESM diffuser was regularly used once a day for solar measurements (and a few monitoring measurements), the ESM mirror is part of all other measurement light paths including regular scientific Earth observations. Thus, the ESM mirror was longer exposed to the harsh UV radiation. For this reason, there was only a very slow build up of the contamination layer on the ESM diffuser. Consequently, the change of the ESM diffuser with time can be neglected.

Due to our approach of not using solar measurements involving the small aperture in the light path, the number of measurements (light paths) in the degradation modelling is limited. Adding the layer thickness on the ESM diffuser as an additional fit parameter does not provide unique/distinct results.

**7. p. 13, lines 3-5:** The magnitude of the SCIAMACHY solar cycle decrease at 330 nm (0.8%) is still somewhat larger than would be expected from TSI change (∼0.1-0.2%). Meanwhile, the 430 nm time series shows an increase of 0.3-0.4% that is out of phase with the TSI variation. This result is discussed more extensively in Section 5.2, but it might be helpful to mention it here.

The focus of this section lies in the comparison of the two versions of SCIAMACHY data. We added the following (bold) here. See also further changes in Sec. 5.2 due to comments of other Referees.

*"With respect to ESA V9.01, a general improvement can be seen in the time series, especially for measurements after 2008. The new results show a more stable signal in the NIR with less small-scale structure. Reasonable results are also obtained in the UV near 330 nm. Here ESA V9.01 shows variations that can not be attributed to natural solar variability, while our results show more clearly the continuous decrease as expected from the descending phase of solar cycle 23 with a clear minimum at 2008/2009 as evident from other SSI measurements (e.g. Mauceri et al., 2018).* ***However, values are higher than expected from TSI variability (∼0.1% solar cycle; Kopp, 2016). Furthermore, an unexpected anti-cyclic increase during solar minimum, similar to the behaviour of V9.01, becomes evident in the NUV and above; see further discussion and comparisons with other SSI data sets in Section 5."***

**8. p. 17, lines 1-2:** I feel that the authors have done a lot of excellent work to reach this level of accuracy.

**9. p. 17, lines 12-14:** The SCIAMACHY SSI data are valuable for studies of short-term

solar variations because they have sufficient spectral resolution in the visible and near-IR to provide unique information about the behavior of solar absorption features in those spectral regions.

**10.  p. 18, lines 24-25:** Are there any plans to release these revised SSI data? Is there a timetable for the creation of an ESA Version 10 data product?

ESA projects to study further SSI from SCIAMACHY have been initiated. At the moment it is not possible to give a schedule for possible data release. It is also still unclear, if the work will be used in the next version, as the priorities of ESA are the impact on level 2 (atmospheric parameters) products.

The data set derived in this study and shown in the paper will be made available at http://www.iup.uni-bremen.de/UVSAT/datasets

**TYPOGRAPHICAL ERRORS**
p. 1, line 22: "simultaneous" should be "simultaneously".
p. 6, line 9: "begin" should be "beginning".
p. 6, line 31: "reasonable" should be "reasonably".
p. 12, line 12: "now clearly the" could be changed to "now clearly follow the".

All typographical errors are corrected.

Please also note the supplement to this comment:
https://www.atmos-meas-tech-discuss.net/amt-2019-433/amt-2019-433-AC1-
supplement.pdf

---

## Author Comment (AC2) · 24 Apr 2020

We thank the reviewer for the detailed remarks and the effort to improve our paper. We responded to all comments as best as we can. Section 5.2 of the manuscript with marked changes is attached to the answers.

**In general, this is a good paper and I recommend only a few minor corrections.**

**Section 3.4 is about the aging correction for the WLS. As I understand your text, the WLS is only used to make a flat-field correction to the detector array.**

**Corrections to the long-term trends in the SSI time series come from the other sources. Is this correct?**

In the operational processor, the original purpose of the WLS measurements was to derive corrections for several effects: e.g. verification of the in-flight memory effect, the pixel-to-pixel gain, and the etalon effect. Within our project, which is focussing on the solar measurements by SCIAMACHY, the on-ground to in-flight correction used the WLS measurements. It corrects for all calibration changes until a reference day shortly after launch (Hilbig et al., 2018). As a stable light source the WLS was intended for long-term monitoring of the instrument. In this study/paper, it is used for the first time to derive a degradation correction; or rather we added the time series of WLS measurements in the fit of the degradation parameter.

**In the new degradation model, the WLS is an "independent" light source. The corrections shown in Figure 5 and described in Equation 1 seem to assume that all of the observed changes in the lamp are due to degradation of the lamp and none of the loss of signal is due to the rest of the system. Fitting a curve to remove the variation in the WLS seems to make an assumption about the magnitude of changes in the detector (for example). If I have interpreted the text correctly, can you add some discussion on how this changes the trends in the SSI time series? If I have not interpreted the text correctly, then could you add another paragraph explaining how the WLS correction does not impact the final degradation correction?**

The WLS data (as shown in Fig. 5) include a degradation correction: It is shown for V9.01 in the upper panel of Fig. 5 and with our newly derived degradation correction in the lower panel. In the upper panel (V9.01) deviations from a constant signal are observed. This is caused by a combination of uncorrected instrument degradation

and WLS ageing. We know from the literature e.g. Sperling et al. (1996) that the lamp ageing follows (approximately) an exponential curve. In the first step of our new approach we derive and correct only the WLS ageing by fitting an exponential curve. The ageing corrected WLS time series (green line in Fig.5) show remaining deviations from a constant signal. This is attributed to uncorrected instrument degradation and is addressed in the second step of our approach. After we got a new instrument degradation correction, we derived again the WLS ageing correction and so on. With this iterative approach we can improve and better separate both the WLS ageing and the instrument degradation correction. This iterative approach is described in detail in the following section 3.5 and a sketch of the iterative approach is shown in Fig.6.

**In Figure 9, the 430 nm time series shows out of phase trends. Is this a statistically significant result about the Sun, or does this fall within the uncertainty of your SSI time series? In Figure 10, the 425-435 band also shows this behaviour. In the text, you do mention the out of phase behaviour in the SCIAMACHY data, but you don't make a clear statement on whether or not this is a new finding of SCIAMACHY. I would like to see a clarification on this point.**

Currently, this algorithm is not mature enough to distinguish instrumental from physical trends. Therefore, the out of band trends at 430 nm falls within the uncertainty of the correction method. We add two sentences to the paper:

At Sec. 4, p13: *". . . an unexpected anti-cyclic increase during solar minimum, similar to the behaviour of V9.01, becomes evident in the NUV and above; see further discussion and comparisons with other SSI data sets in Section 5."*

At Sec. 5.2, p16/17: *"The observed anti-correlation for the new SCIAMACHY results is therefore likely a remaining residual instrument artefact."*

[Figure]

**P16 L12: you refer to Woods' MuSIL and Mauceri's SIMc as if they were obser-
vational datasets. Both are essentially corrections based on proxies (or TSI)
rather than instrument data. So comparisons to those time series should fall in
the same category as comparisons to SATIRE-S or NRLSSI2.**

Thank you for clarification. We changed the text as follows: *"Recent studies by
Woods et al. (2018) and Mauceri et al. (2018) developed new methods to account
for uncorrected degradation in SIM SSI. Both results, the MuSIL-corrected SIM and
SIMc, show better agreement with independent SSI data such as SATIRE-S than the
operational SIM product (Harder, 2019)."*

**Figure 11 uses different colors for the different missions than Figure 9. I would
recommend that each instrument in the two figures have a consistent color
assigned. It will make it easier for the reader to compare the time series.**

The Figure 11 is updated, so that each instrument has consistent colour assigned in
all figures.

Please also note the supplement to this comment:
https://www.atmos-meas-tech-discuss.net/amt-2019-433/amt-2019-433-AC2-
supplement.pdf

---

## Author Comment (AC3) · 24 Apr 2020

We thank the reviewer for the detailed remarks and the effort to improve our paper. We responded to all comments as best as we can. We attached Section 5.2 (comparisons of data sets) of the changed manuscript to the answers.

**General comments:**

This paper deals with new progress about the instrumental corrections of the SCIA-MACHY data. The special interest is to improve the solar spectral data. The paper is well written and contains interesting and important results for users of SCIAMACHY

data. I would like to recommend the publication of this paper in AMT. I have the following minor comments and questions.

**Specific comments:**

1. p2, line 27: "Shortly after SCIAMACHY, . . .". Add "launch" after SCIAMACHY.

The sentence is changed according to the suggestion: "Shortly after SCIAMACHY was launched, ..."

**2. p3, line15: Please add some comments about the contamination problem: Is SCIAMACHY's contamination problem worse or better when compared with similar instruments? If worse, what went wrong?**

3. p3, line 15: Please, tell briefly about water and the other contaminants: Are they always on separated surfaces (mirrors, diffusers, detectors)?

Contamination, which depends on the outgassing from the platform, is different for each instrument and platform. Even for very similar instruments this may differ strongly as the outgassing from the satellite and its thermal history plays an important role. The degradation of the GOME-2 instruments on MetOp A B and C, which are in principle identical instruments on identical platforms, is similar but not identical.

Identification of the contaminant species and processes that lead to degradation are difficult or hardly possible by in-flight monitoring, as indicated e.g. in Krijger et al. 2014: "Studies like e.g. the one by Stiegman et al. (1993) on diffusers, show some organic effluent present, but did not allow for the identification of the contaminant. Also, Chommeloux et al. (1998) showed the on-ground degradation as a result of UV or photon radiation, as did Georgiev and Butler (2007) or Fuqua et al. (2004). [...] In summary, most of the early satellites suffered from degradation caused by

outgassing. However, the exact identity of the contaminant causing the Ultraviolet to visible (UV–VIS) degradation remained unclear. McMullin et al. (2002) studied the degradation of SOHO/SEM, and found that they could explain the degradation with a thin layer of carbon forming on the forward aluminium filter. The exact source of the contaminant is unknown, but is suspected to be outgassing of the satellite itself. Schläppi et al. (2010) attempted in situ mass spectrometry with ROSETTA to measure the constituents of their contamination, and found the main contaminants to be water. In addition, organics from the spacecraft structure electronics and insulations were identified. Water was also found in SCIAMACHY, where it was deposited onto the cold detectors (Lichtenberg et al., 2006). In fact, Earth-observing satellites suffer from degradation, both in throughput and in the polarisation and/or scan-angle dependence, such as GOME (Krijger et al., 2005a; Slijkhuis et al., 2006), MODIS (Xiong et al., 2007), VIIRS (Lei et al., 2012), MERIS (Delwart, 2010), SCIAMACHY (Bramstedt et al., 2009), and the two GOME-2 (Lang, 2012) instruments currently in orbit."

Lichtenberg et al., 2006, give some explanation about water/ice in SCIAMACHY: "Shortly after the very first cooling of the detectors, a significant loss of transmission in channel 7 and 8 was discovered. Investigations showed that an ice layer growing on top of the cylindrical lens covering the detectors was responsible. Only channel 7 and 8 are affected because these channels are cooled down to around 145 K while the other channels have temperatures of 200 K or higher. A likely source of the contamination is the carbon fibre supporting structure of ENVISAT itself, since it is known that carbon fibres can accumulate a substantial amount of water. The water contained in the fibres started to gas out once the satellite was in orbit. SCIAMACHY is covered by a double layer of multilayer insulation (MLI) blankets, one from ENVISAT and one from the instrument itself to prevent strong thermal gradients while in orbit. The MLI has a number of venting holes to allow the outgassing of the Instrument and prevent a contamination of surfaces, but apparently the venting volume allowed by the

СЗ

holes is not large enough or the holes are obstructed. Thus, the contaminant is not (or too slowly) removed from the instrument volume. Other instruments on ENVISAT have also reported problems due to contamination (see e.g. Perron, 2004; Smith, 2002)."

In the manuscript, we clarified the impact of water for SCIAMACHY. We change:

"Another critical species of contaminant is water that can be deposited on the detectors as was the case with SCIAMACHY (Lichtenberg et al., 2006)."

to:

"Another critical species of contaminant is water that can be deposited on cooled surfaces in the instrument as it was the case for SCIAMACHY's NIR detectors (Lichtenberg et al., 2006). In addition, water vapour is photolysed in the hard UV to generate OH and H free radicals. In particular OH is a very strong oxidising agent and initiates the oxidation of volatile organic compounds which are also photolysed in the UV and result in additional low volatile organic compounds deposited on the optical surfaces of the instrument."

We added this statement to the end of the paragraph (p.3):

"Detailed discussions on contamination of optical surfaces in various space instruments are provided by BenMoussa et al. (2013); Krijger et al. (2014); Meftah et al (2017) among others. Despite the numerous studies available, the composition of contaminants as well as the exact processes for their build-up remain highly uncertain. For each instrument and platform, the individual construction (platform, instrument) and performance lead to different effects, and are difficult to quantify without having direct access in space. SCIAMACHY and its precursor GOME show moderate degradation in the UV and visible spectral range, whereas the successor GOME-2 series show more rapid degradation."

**4. p3: Perhaps you could say something about occultation instruments, which are to some extent more resilient to contamination problems due to the self-calibrating property.**

Occultation instruments are self-calibrating only in case of atmospheric measurements, because then the same type of measurements above and through the atmosphere are divided to derive the transmission spectra. An absolute radiometric calibration of irradiance measurements is therefore not required ("self-calibrating"). As far as we are aware of, occultation instruments usually do not provide absolute calibrated solar spectra. Even SCIAMACHY's solar occultation measurements do not provide calibrated irradiance. Therefore we think occultation measurements are not relevant for this paper.

**5. p7, line 10: Another kind of. . . . Is this story part of the story on line 13 where you tell that "there are basically two types. . . "? I am confused.**

The paragraph summarises the issues of the selected light paths (measurements) in the V9.01 degradation correction. For some light paths several issues arise:

- missing sub-solar measurements 2002
- changed temperature settings (all light paths)
- measurements including small extra mirror
- small aperture (sub-solar, limb, extra-mirror light paths)

The following paragraph (in Sec. 3.2.) was changed from

"The following issues arise in the current degradation correction. Firstly, the sub-solar

measurements were missing in 2002 and therefore the OBM m-factors before February 2003 were calculated assuming fixed settings for the contamination layers (as defined at the reference day in February 2003) and a constant Sun for the ESM diffuser light path. Additionally, temperature settings of the instrument were changed in February 2003. **Another kind of solar measurements** uses an extra mirror such that the light path crosses the ESM mirror twice. The extra mirror ..."

to

"The following issues arise in the current degradation correction. Firstly, for operational reasons, the sub-solar measurements were not made in 2002. As a result the OBM m-factors before February 2003 were calculated assuming fixed settings for the contamination layers (as defined at the reference day in February 2003) and a constant Sun for the ESM diffuser light path. Additionally, temperature settings of the instrument were changed in February 2003. An additional solar light path uses an extra mirror such that the light crosses the ESM mirror twice. The extra mirror..."

**6. p7, line 25: You have detected spectral features by the ESM diffuser. On line 26 you say that's why the ASM diffuser was added but without pre-launch calibration. Are you sure that ASM does not have similar spectral features than ESM?**

The pre-flight calibration identified spectral features in the ESM diffuser. As a result an ASM diffuser was added to the instrument. As mentioned in the text, an absolute radiometric calibration was not possible, because the ASM diffuser was added to the instrument after the (commercial) radiometric calibration campaign was completed. Pre-launch characterisation was undertaken and showed that the ASM diffuser has much less spectral features. The features in the ESM have two possible origins: i) the inability to make truly amorphous diffuser plates by the manufacturer and ii) the ring used to hold the ESM diffuser in place may have also introduced some optical effects. The ASM diffuser is significantly larger than the ESM diffuser and have much less residual parallel scattering, which induces etalon features.

**7. p12, line 5: I get the impression that the ASM diffuser contamination increases along the mission, but the ESM contamination remains constant. Is this right and what is the reason?**

The contamination layer thickness of the ESM Diffuser is kept constant in the modified degradation correction. This assumption is based on previous studies for SCIAMACHY V8/9. The ASM Diffuser was included in the model for the first time within this study, and therefore, no previous values exist. Overall our results indicate that the ASM Diffuser has a relatively small and constant contamination during the mission. We assume that the cause of contamination is the accumulated exposure to solar UV radiation. This is confirmed by previous investigations of the SCIAMACHY degradation. See also answer to Referee #1, item 6.

**8. p14, line 5: Could you tell something more about the change of the thermal environment?**

Gottwald et al, 2016, SCIAMACHY In-orbit Mission Report, p. 67, reported: "Shortly before the ENVISAT orbit manoeuvre in October 2010 an anomaly occurred in the Ka-band antenna subsystem (KBS). This required switching from KBS-2 to KBS-3 and to a change in its operating procedure. While KBS-2 had been intermittently turned 'on' and 'off', for safety reasons, KBS-3 remained 'on' the whole time. Therefore about 120 W more energy were dissipated thus changing the thermal environment of ENVISAT, including the payload instruments." Effects on SCIAMACHY were e.g.

increased detector temperatures (0.3 K to 0.5 K, channel dependent) and increased Electronic Assembly subsystem temperatures (1.1 to 2.7 °C, subsystem dependent).

In the text (Sec. 4), we changed:

"A possible explanation is a change in the use of electrical devices on Envisat by the end of October 2010 that resulted in a change in the thermal environment (Gottwald et al., 2016)."

to:

"A possible explanation is the additional permanent use of a backup system after an anomaly in the Ka-band antenna subsystem. About 120W more energy were dissipated resulting in a change of the thermal environment of Envisat. Increased temperatures ( $0.3 - 2.7 \degree C$ , subsytem dependent) were observed by SCIAMACHY's internal temperature sensors (Gottwald et al, 2016). This changes the outgassing and thus the photochemical processing."

**9. p15, line 22: outlier ->outliers** Corrected.

**10. p16, line 5: At 330nm in Fig. 10 the SIM curve seems to deviate clearly from the other curves, but your comment is more positive. Could you explain?**

In this part of the paper the intention was not to explain differences but rather to describe them. Since our new SCIAMACHY SSI result also deviates from the other data sets, detailed comparisons are not applicable at this stage. We expanded and reordered the paragraph. Please see changed manuscript.

11. p18, Conclusions: Your conclusions are quite upbeat which is understandable after very tedious and extensive work. But if I consider the results shown in Fig.10 to be the most important outcome, the situation is not looking very promising. Is it possible that something important is still waiting to be found?

The intention of the paper was to present the method and the successful implementation of the WLS and ASM diffuser solar measurements in the model. We agree, Figure 10 shows indeed some weakness in the SCIAMACHY SSI with respect to other instrument or model data. Nevertheless, the improvement over the latest (unpublished) SCIAMACHY V9.01 is demonstrated. Comparisons with older SCIAMACHY version are not possible, since V9.01 is the first one that accounts for solar variability on solar cycle time scales. The method has the potential to produce improved results for SCIAMACHY after additional modifications. In addition the paper demonstrates the value of using WLS observations for degradation monitoring for other instruments, as mentioned in the conclusions.

Nevertheless, we change in the conclusion:

"there are still some limitations in the degradation corrections"

to

"there are still limitations in the degradation corrections" (omitting "some")

Please also note the supplement to this comment: https://www.atmos-meas-tech-discuss.net/amt-2019-433/amt-2019-433-AC3supplement.pdf